# ERp29 Attenuates Nicotine-Induced Endoplasmic Reticulum Stress and Inhibits Choroidal Neovascularization

**DOI:** 10.3390/ijms242115523

**Published:** 2023-10-24

**Authors:** Tu Lu, Fangfang Xie, Chuangxin Huang, Lijun Zhou, Kunbei Lai, Yajun Gong, Zijing Li, Longhui Li, Jiandong Liang, Qifeng Cong, Weihua Li, Rong Ju, Sarah X. Zhang, Chenjin Jin

**Affiliations:** 1State Key Laboratory of Ophthalmology, Zhongshan Ophthalmic Center, Sun Yat-sen University, Guangdong Provincial Key Laboratory of Ophthalmology and Visual Science, Guangzhou 510060, China; 2Department of Ophthalmology and Ross Eye Institute, University at Buffalo, State University of New York, Buffalo, NY 14203, USA; 3SUNY Eye Institute, State University of New York, Buffalo, NY 14203, USA; 4Department of Biochemistry, University at Buffalo, State University of New York, Buffalo, NY 14203, USA

**Keywords:** ERp29, nicotine, endoplasmic reticulum stress, macrophage polarization, choroidal neovascularization

## Abstract

Nicotine-induced endoplasmic reticulum (ER) stress in retinal pigment epithelium (RPE) cells is thought to be one pathological mechanism underlying age-related macular degeneration (AMD). ERp29 attenuates tobacco extract-induced ER stress and mitigates tight junction damage in RPE cells. Herein, we aimed to further investigate the role of ERp29 in nicotine-induced ER stress and choroidal neovascularization (CNV). We found that the expression of ERp29 and GRP78 in ARPE-19 cells was increased in response to nicotine exposure. Overexpression of ERp29 decreased the levels of GRP78 and the C/EBP homologous protein (CHOP). Knockdown of ERp29 increased the levels of GRP78 and CHOP while reducing the viability of ARPE-19 cells under nicotine exposure conditions. In the ARPE-19 cell/macrophage coculture system, overexpression of ERp29 decreased the levels of M2 markers and increased the levels of M1 markers. The viability, migration and tube formation of human umbilical vein endothelial cells (HUVECs) were inhibited by conditioned medium from the ERp29-overexpressing group. Moreover, overexpression of ERp29 inhibits the activity and growth of CNV in mice exposed to nicotine in vivo. Taken together, our results revealed that ERp29 attenuated nicotine-induced ER stress, regulated macrophage polarization and inhibited CNV.

## 1. Introduction

Age-related macular degeneration (AMD) is a common condition worldwide [1] and is one of the main causes of irreversible blindness [2,3]. Wet AMD is mainly characterized by choroidal neovascularization (CNV) [4,5]. It has been shown that 90% of severe visual impairment in patients with AMD is caused by CNV [6]. While anti-vascular endothelial growth factor (VEGF) treatments can help control or improve CNV, some patients still develop atrophic lesions after treatment. Therefore, further research is needed to understand the mechanisms underlying wet AMD and explore new treatment strategies [7,8]. Additionally, some studies have reported insensitivity and drug resistance to anti-VEGF treatment [8,9,10].

Among various risk factors, which mainly include genetic susceptibility and environmental exposure, smoking plays an important role in the development of AMD [3,11]. The risk of AMD among people who smoke is twice as high as that in the normal population, and the onset of AMD occurs 10 years earlier in patients who smoke [12]. Previously, Huang CX et al. [13,14] showed that tobacco extract induces endoplasmic reticulum (ER) stress and oxidative stress in retinal pigment epithelium (RPE) cells, exacerbates the unfolded protein response and impairs RPE tight junctions. These findings suggest that smoking is an early biological event that contributes to the progression of AMD. Moreover, endoplasmic reticulum protein 29 (ERp29) [15], a novel chaperone whose expression is decreased in an AMD mouse model [16,17], attenuates ER stress and mitigates tight junction damage in RPE cells [14].

Nicotine is one of the most important toxic and harmful components of tobacco, and it is highly addictive [18,19]. In vivo studies have shown that long-term exposure to nicotine can increase the severity of laser-induced CNV in mice by regulating macrophage polarization [20]. On the other hand, nicotine exposure can promote RPE cell secretion of macrophage-related chemokines, such as fibronectin, laminin, and monocyte chemoattractant protein [21], to recruit and activate macrophages. These macrophages then infiltrate the inflammatory microenvironment of the retinal RPE, further contributing to the formation and progression of CNV [22]. Nicotine also induces M2 macrophage polarization through specific macrophage receptors, and M2 macrophages then secrete a series of inflammatory and angiogenic factors into the inflammatory microenvironment, thereby assisting the formation and development of CNV [23]. RPE cells and macrophages are the main sources of VEGF secretion in CNV lesions [24,25], and both play crucial roles in the formation of CNV [26]. By studying the interaction between RPE cells and macrophages, we can gain better insights into how nicotine exposure exacerbates CNV.

The purpose of this study was to (1) observe the changes in the levels of ER chaperone proteins in RPE cells after nicotine exposure; (2) elucidate the function of ERp29 in RPE cells under conditions of nicotine-induced ER stress; and (3) further observe the role of ERp29 in changing the interaction among RPE cells, macrophages, and vascular endothelial cells in the CNV microenvironment.

## 2. Results

### 2.1. Nicotine Exacerbates ER Stress and Upregulates the Expression of ERp29 in ARPE-19 Cells

Previously, we showed that ERp29 attenuates cigarette smoke extract (CSE)-induced ER stress, reduces apoptosis, and mitigates tight junction damage in RPE cells [14]. It is not clear whether nicotine, which is one of the main components of cigarette extract, has the same effect on RPE cells. To investigate whether nicotine affects ER stress, ARPE-19 cells were exposed to different concentrations of nicotine for up to 48 h. As shown in Figure 1A, nicotine exposure increased the expression of ERp29 (Figure 1B) and GRP78 (Figure 1C), and the effect was the most significant when cells were exposed to 20 μM nicotine for 24 h. As shown in Figure 1D, nicotine exposure increased the expression of ERp29 (Figure 1E) and GRP78 (Figure 1F), and the effect was the most significant at 24 h or 48 h. These results suggest that nicotine induces ER stress and upregulates the expression of ERP29 and GRP78 in ARPE-19 cells. Nicotine had no apparent effect on cell viability under the experimental conditions described above but did affect cell viability at a concentration of 2500 μM for 24 h (Figure 1G,H).

### 2.2. ERp29 Attenuates Nicotine-Induced ER Stress and Protects ARPE-19 Cells

Our previous study showed that ERp29 can reduce CSE-induced ER stress by modulating the expression of ER chaperones and other ER stress-inducible proteins [14]. To determine whether ERp29 regulates nicotine-induced ER stress in ARPE-19 cells, we overexpressed ERp29 via lentivirus. Cells transduced with negative control lentivirus (LV-NC) were used as a control. Immunostaining showed that ERp29 was stably overexpressed in the cells after lentivirus transfection. Real-time qPCR and Western blotting assays indicated that the expression of ERP29 was significantly increased compared with that in the negative control group. After lentivirus transduction, the cells were exposed to 20 μM nicotine for 24 h, and Western blotting analysis was performed to examine the expression of target proteins. The Western blotting results showed that overexpression of ERp29 significantly decreased the levels of GRP78 (Figure 2B) and the C/EBP homologous protein (CHOP, Figure 2D) in response to nicotine exposure. To determine whether ERp29 protects ARPE-19 cells from nicotine exposure, a CCK-8 assay was performed (Figure 2E,F). More cells in the ERp29 group survived nicotine exposure than in the negative control group.

### 2.3. Knockdown of ERp29 Expression Exacerbates Nicotine-Induced ER Stress and Decreases the Viability of ARPE-19 Cells

To further understand the role of endogenous ERp29 in regulating ER stress, ERp29 siRNA was used to downregulate the expression of ERp29. After transfection into ARPE-19 cells, siR-ERp29 effectively suppressed the expression of ERp29 at the mRNA and protein levels compared with siR-NC. Then, the cells were transfected with siR-ERp29 siRNA or negative siRNA for 48 h and treated with 20 μM nicotine. Western blotting assays showed that knockdown of ERp29 significantly upregulated the expression levels of GRP78 and CHOP compared with those in the nicotine-induced group. The Western blotting results suggested that ERp29-deficient cells may be sensitive and vulnerable to ER stress-related cell death (Figure 3B,D). To further determine whether downregulation of ERp29 affects the survival of nicotine-treated ARPE-19 cells, we examined the viability of ARPE-19 cells transfected with siR-ERp29 or siR-NC. The CCK-8 assay showed that cytotoxic effects of nicotine were increased in ERp29-deficient cells (Figure 3E,F).

### 2.4. ERp29 Regulates Macrophage Polarization In Vitro

ER stress in RPE cells plays an important role in CNV formation. Furthermore, nicotine can exacerbate CNV by promoting M2 polarization. We hypothesized that ERp29-overexpressing RPE cells could affect macrophage polarization in an inflammatory microenvironment. To further elucidate the mechanism by which ERp29 affects CNV formation, we established ARPE19 cell/M0 macrophage and ARPE19 cell/M2 macrophage coculture systems. First, to generate a model of macrophage polarization, we stimulated PMA-differentiated human THP-1 monocytes (M0 macrophages) with IL-4 and IL-13 (M2 polarized macrophages). THP-1 monocytes cultured in the presence of PMA became adherent and round, and real-time qPCR was used to measure the expression of known M0 and M2 macrophage markers. As expected, the mRNA expression of the M0 macrophage markers CD11b and CD14 was increased, and the mRNA expression of the M2 macrophage markers CD163, CD206, IL-10, and CCL17 was also increased. These results suggested the successful polarization of THP-1 monocytes into M0- and M2-polarized macrophages. Then, the macrophages were used to establish ARPE19 cell/M0 macrophage and ARPE19 cell/M2 macrophage coculture systems. As shown in Figure 4A–C, compared with the LvNC group, the LvERp29 group in the coculture system significantly reduced the expression of CD163, CD20, and IL-10 (Figure 4A–C) in macrophages and increased the expression of iNOS and CCR7 (Figure 4D,F), indicating that ERp29 effectively inhibits M2 macrophage polarization in vitro.

### 2.5. ERp29 Inhibited HUVECs Proliferation, Migration, and Tube Formation In Vitro

As mentioned in the previous section, ERp29-overexpressing RPE cells could affect macrophage polarization in the inflammatory microenvironment. We extracted conditioned media from the ARPE-19 cell/macrophage coculture systems to observe how ERp29 affects angiogenesis. The conditioned media were added to human umbilical vein endothelial cell (HUVEC) cultures, and their effect on the cell viability, migration, and tube formation of HUVECs was observed. For the CCK-8 assay, the results showed that compared with that from the LvNC group, the condition medium from the LvERp29 overexpression group inhibited the proliferation of HUVECs (Figure 5A,B). In the wound-healing assay, the results showed that the migration rate in the LvERp29 group was significantly decreased compared to that in the LvNC group (Figure 5C,D). In the tube formation assay, the results showed that overexpression of ERp29 reduced the length of the vessels compared to that in the LvNC group (Figure 5E,F).

### 2.6. ERp29 Inhibits the Activity and Growth of Mouse CNV In Vivo

To explore the effect of ERp29 in vivo, we established a laser-induced mouse CNV model and provided these mice with water supplemented with nicotine. We used adeno-associated virus (AAV) to construct the ERp29 overexpression vector and negative control. C57BL/6J mice were intravitreally injected with AAV, and three weeks later, laser-induced CNV was observed. Seven days after the model was established, we compared CNV activity and size between the AAV-ERp29 group and the AAV-NC group (Figure 6A). The results demonstrated that the AAV-ERp29 group had less vascular leakage than the AAV-NC group in both the early phase and late phase of FFA. The histogram shows the leakage scores and grade distribution of each group (Figure 6B–D). The AAV-ERp29 group had smaller CNV area and volume than the AAV-NC group, as shown by immunostaining for IB4 (Figure 6E–G).

## 3. Discussion

In the current study, we present in vitro evidence that nicotine exposure exacerbates ER stress in RPE cells. Furthermore, ERp29 attenuates ER stress and protects RPE cells from the consequences of nicotine exposure. To observe how ERp29 in RPE cells affects macrophages in vitro, we established an RPE cell/macrophage coculture system. We found that ERp29 indirectly regulates macrophages by inhibiting M2 polarization (Figure 4). Vascular endothelial cell functions, such as viability, migration, and tube formation, were inhibited by conditioned medium from ERp29-overexpressing RPE cells (Figure 5). Our in vivo observations showed that ERp29 inhibits CNV formation and activity (Figure 6).

ERp29 is a member of the protein disulfide isomerase chaperone family of endoplasmic reticulum luminal proteins with different domains, and it performs two main biological functions [15]. First, ERp29, which is an ER chaperone, together with GRP94, GRP78, ERp72, calnexin and other molecules, regulates the level of unfolded proteins and ER stress [27]. Second, ERp29 is an escort protein that is involved in COP vesicle trafficking [28,29,30,31]. ERp29 is involved in the process of protein transport between the ER and Golgi apparatus or the process of protein transport to secretory vesicles [32,33,34]. Both of these functions are important for regulating ER stress and helping cells cope with stress in their microenvironment. Previous studies have revealed that ERp29 is a marker of ER stress disturbance in RPE cells that are exposed to tobacco extract [14]. ERp29 and GRP78 may play similar roles in maintaining endoplasmic reticulum homeostasis [35]. Our nicotine study shows that ERp29 may also be a molecular marker that can indicate endoplasmic reticulum stress homeostasis. In addition, in a study on liver injury and regeneration, Cao XY et al. found that during liver regeneration, the expression of ERp29 increases to regulate the level of endoplasmic reticulum stress and repair the regeneration of injured tissues [36]. These findings suggest that ERp29 is a potential target for protecting cells during the progression of retinal and neurodegenerative disease [37].

RPE cells are responsible for maintaining the normal function and regulating the immune microenvironment of the retina. It is necessary for RPE cells to maintain the homeostasis and integrity of tight junctions. Additionally, RPE cells have a direct impact on the function of immune cells by controlling the inflammatory microenvironment in the retina. First, RPE cells secrete cytokines to induce and enhance the regulatory activity of Treg cells [38,39]. Second, RPE cells can also secrete soluble cytokines to inhibit the activation of effector T cells [40,41]. Third, RPE cells have a potential ability to regulate antigen-presenting cells. Furthermore, RPE cells regulate the initial activation of T cells and presentation of retinal antigens in regional lymph nodes by inhibiting dendritic cell activation [42,43]. Previous studies have shown that conditioned medium extracted from RPE eye cups can suppress the production and secretion of proinflammatory cytokines by macrophages. The factors that are secreted by RPE cells can target macrophages and microglia to enhance their anti-inflammatory abilities and activate Treg cells. [44,45]. In conclusion, RPE cells are thought to play a critical role in regulating the immune system within their physiological microenvironment.

In the microenvironment of CNV, macrophage-mediated inflammation plays a role in neovascularization. RPE cells, which endure long-term stress, can recruit macrophages to the inflammatory microenvironment through cellular interactions and then regulate their activity and polarization. When ERp29 assists in maintaining homeostasis, RPE cells regulate the factors they secreted through paracrine signaling, which helps ameliorate the inflammatory microenvironment. In addition, epithelial cells can also interact with other cells in the microenvironment by secreting exosomes [46,47,48], thereby disrupting or maintaining the homeostasis of the CNV microenvironment. The proteins and RNA that are carried by exosomes derived from RPE cells under stress may also participate in the regulatory process of macrophages [47], but further research is needed to understand this process.

Neovascularization is related to the infiltration of macrophages into the inflammatory microenvironment [49]. Zhang H et al. found that M2 macrophages actively participate in the process of tumor growth. Within the tumor microenvironment, Wu KY et al. defined M2 macrophages as tumor-promoting macrophages and M1 macrophages as antitumor macrophages [50]. The growth of CNV (choroidal neovascularization) also depends on the involvement of macrophages. Through the observation of a laser-induced CNV model, relevant studies have shown that ICAM-1 mediates the chemotaxis of F4/80+ macrophages, which then reside in the inflammatory microenvironment of CNV lesions. These macrophages promote CNV growth by upregulating VEGF, TNF-α, and MMP-2 [22]. Another team also revealed that M2 polarization induces the formation and exacerbation of CNV [51,52].

Our findings indicate that when ERp29 is overexpressed, the resulting conditioned medium exerts indirect inhibitory effects on the growth, migration, and tube formation of vascular endothelial cells. Additionally, RPE cells overexpressing ERp29 can resist the effects of nicotine, alleviate ER stress, and restore cell homeostasis. It is possible that the regulatory effect of ERp29 on macrophage polarization may inhibit the function of vascular endothelial cells or ERp29 may directly affect vascular endothelial cell function. Li WD et al. [53] showed that P300 regulates the endoplasmic reticulum stress molecule XBP1s, which in turn controls M2 macrophage polarization and ultimately influences the growth of CNV. This finding supports the idea that macrophages can be influenced by regulating cellular homeostasis in the microenvironment, which subsequently affects the development of neovascularization. Numerous studies have been published on mechanisms to inhibit macrophage M2 polarization, including the use of hormones such as melatonin [54], herbal extracts such as triptolide [55,56], and microRNAs [57,58]. Molecular drugs with both anti-inflammatory and anti-VEGF effects may offer a new approach for the treatment of CNV [58].

To further study the function of ERp29 in vivo, we need to use RPE-CRE mice to observe changes in the retina when ERp29 is knocked out in RPE cells [59]. ERp29 deficiency mimics the pathological process of retinal neurodegeneration that is associated with the disruption of RPE cell homeostasis, and this may be helpful for further understanding AMD. We can also examine the factors that are secreted by RPE cells in the homeostasis disruption model. However, there are some problems that need our attention. Although a high level of ERp29 expression can inhibit CNV in vivo, the effect of its overexpression on other cells in the eye as well as its safety with respect to visual function still require further research. In addition, ERp29 is highly expressed in various tumor tissues, and some researchers have identified ERp29 as a carcinogenesis-related gene [60,61]. In addition to the upregulation of ERp29 expression in tumor tissues, the expression of ERp29 can also be upregulated after ionizing radiation [62,63]. In the field of antitumor drugs, ERp29 was upregulated in a p53-dependent manner in PC3 prostate cancer cells and mouse embryonic fibroblasts after treatment with antitumor chemotherapy drugs [63]. Upregulation was also observed in thyroid cells and INS1 pancreas β cells after drug treatment [64,65]. Interestingly, compared with that in resting mammary glands, the expression of ERp29 in lactating mammary gland cells was also upregulated [66]. Although the high expression of this cellular stabilization protein may exert neuroprotective effects, the risks of promoting carcinogenesis still need further observation. The relationship between ERp29 and cancer should also be more thoroughly analyzed.

## 4. Materials and Methods

### 4.1. Cell Culture and Cell Treatment

The human ARPE-19 RPE cell line and the human THP-1 monocyte cell line were purchased from Procell Life Science & Technology (Wuhan, China). Human umbilical vein endothelial cells were purchased from the American Type Culture Collection (ATCC, Mansas, VA, USA).

### 4.2. Construction and Transduction of Lentivirus

Recombinant lentivirus encoding human ERp29 (pSLenti-EF1-EGFP-P2A-Puro-CMV-ERp29-3xFLAG-WPRE) and lentivirus encoding a negative control (pSLenti-EF1-EGFP-P2A-Puro- CMV-3xFLAG-WPRE) were purchased from OBiO Technology Company Ltd. (Shanghai, China). ARPE-19 cells were transfected with virus using polybrene (OBiO, Shanghai, China) according to the manufacturer’s recommendations. After 72 h of transfection, ARPE-19 cells were observed with a fluorescence microscope to confirm that the lentivirus had been successfully transduced into the target cells.

### 4.3. Small-Interfering RNAs (siRNAs)

Three different siRNA-ERp29 sequences were designed and synthesized by RiboBio Company (Guangzhou, China), and these sequences specifically bound to and facilitated the degradation of ERp29 mRNA. Using the RiboFect cP transfection kit (RiboBio, Guangzhou, China), ARPE19 cells were transfected with siRNA-ERp29-1, siRNA-ERp29-2, and siRNA-ERp29-3 (100 nM) in 6-well plates for 24 h. After transfection, Western blotting and PCR were performed to select the most efficient sequence. The sequences of the three siRNAs were as follows: siRNA-ERp29-1, 5′-TGGATACGGTCACTTTCTA-3′; siRNA-ERp29-2, 5′-CCCTGGATACGGTCACTTT-3′; and siRNA-ERp29-3, 5′-GATACGGTCACTTTCTACA-3′.

### 4.4. Cell Viability and Proliferation Assays

Previously collected coculture media were used to prepare conditioned media. Briefly, the media were centrifuged at 400× *g* for 10 min to remove whole cells, and the supernatants were centrifuged again at 5000× *g* for 20 min to remove cell debris. After that, the supernatant was filtered through a 0.22 μm filter. HUVECs were treated with this conditioned media, and their proliferation was measured with the CCK-8 assay (Vazyme, Nanjing, China). Cells were seeded in 96-well plates (5 × 10^3^ cells/well) for 24 h, the medium was replaced with conditioned medium, and the cells were cultured for another 24 h or 48 h. Finally, 10 μL of CCK8 working solution was added to each well and incubated at 37 °C. The absorbance was measured at 450 nm using a microplate reader (BioTek, Winooski, VT, USA).

### 4.5. Western Blotting Analysis

Total protein was extracted using RIPA Lysis Buffer (Beyotime, Shanghai, China) containing a protease inhibitor cocktail on ice for 20 min. According to the manufacturer’s instructions, the BCA protein assay kit (Beyotime, Shanghai, China) was used to determine the protein concentration. Proteins were separated by 10% SDS–PAGE and then transferred to 0.45 µm polyvinylidene difluoride membranes (Millipore, MA, USA). After blocking each membrane with 5% nonfat milk for 2 h at room temperature, the membranes were incubated with primary antibodies at 4 °C overnight. After washing three times with Tris-buffered saline-0.1% Tween-20 (TBST), the membranes were incubated with secondary antibodies at room temperature for 1 h. The membranes were again washed three times with TBST and then visualized using an Immobilon Western Chemiluminescent HRP Substrate (Millipore, MA, USA). The antibodies are listed in Appendix A.

### 4.6. RNA Isolation and Quantitative PCR (qPCR)

Total RNA was extracted from cells using the TRIzol reagent. The concentrations and purity of the RNA were determined with the NanoDrop 2000 (Thermo Fisher, Waltham, MA, USA). Approximately 1 μg of total RNA was reverse transcribed into cDNA using HiScript III RT SuperMix (Vazyme, Nanjing, China), and real-time qPCR was performed using the Taq Pro Universal SYBR qPCR Master Mix (Vazyme, Nanjing, China) on an Applied Biosystems QuantStudio 6&7 according to the manufacturer’s protocol. The PCR conditions were as follows: initial denaturation at 95 °C for 30 s, 40 cycles of PCR followed by 95 °C for 5 s, 58 °C for 30 s, and 72 °C for 1 min. The 2^−ΔΔcq^ method was used to analyze the relative expression of different genes. The primer sequences are shown in Appendix A.

### 4.7. Macrophage Polarization

THP-1 cells were differentiated into M0 macrophages by incubation with 320 nmol/L PMA for 24 h. To obtain M2 macrophages, THP-1 cells were treated with 320 nmol/L PMA for 12 h and then incubated with 100 nmol/L PMA plus 20 ng/mL IL-4 and 20 ng/mL IL-13 for an additional 48 h.

### 4.8. ARPE-19 Cell/M2 Macrophage Coculture System

To further understand how ERp29 inhibits angiogenesis, Transwell polycarbonate membrane cell culture inserts (Boyden chamber, Corning, NY, USA) were used to establish a ARPE19 cell/M2 macrophage coculture system. Briefly, THP-1 cells were seeded on the bottom of each well of 6-well plates. Transwell inserts were placed into the wells above the THP-1 cells. According to the different experimental groups, THP-1 cells were differentiated into M0 macrophages or M2 macrophages by incubation with the supplements described above. ARPE19 cells that were transfected with lentivirus were seeded in the inserts. Transwell coculture plates were then incubated at 37 °C in 5% CO_2_ for 48 h. Then, the media were collected from the Transwell coculture 6-well plates to prepare conditioned media for further study. The cells were collected from the 6-well plates, and total RNA was extracted.

### 4.9. Wound-Healing Assay

Culture-Insert 2 wells were used for the wound-healing assay. Culture inserts were first inserted into the center of each well of the 6-well plate, and then HUVECs were seeded in the 6-well plates (2 × 10^6^ cells/well). After the cells were attached, the inserts were removed, and a scratch was generated in the cell monolayer. The cells were then cultured in conditioned medium from different groups for 6–24 h. Cell migration was recorded with an inverted microscope (Olympus, Tokyo, Japan).

### 4.10. Tube Formation Assay

Two hundred microliters of basement membrane substrate (Corning, NY, USA) was used to coat the bottom of each well of 24-well plates. The plates were incubated for 30 min at 37 °C to allow a gel layer to form. Then, HUVECs were seeded in each well (5 × 10^4^ cells/well) and incubated with conditioned medium from different groups for 24 h. Finally, an inverted microscope was used (10× objective) to capture images of four randomly selected areas around the center of the formed lumen.

### 4.11. Mouse Model of CNV and Administration

Male C57BL/6J mice (6–8 weeks old, SPF grade) were purchased from the Animal Laboratory of the State Key Laboratory of Ophthalmology, Zhongshan Ophthalmic Center, Sun Yat-sen University (Guangzhou, China). The animal experimental and care procedures were approved by the Institutional Animal Care and Use Committee of Zhongshan Ophthalmic Center and complied with the ARVO Statement for the Use of Animals in Ophthalmic and Vision Research.

Mice were given water with or without nicotine in a light-proof bottle, and the water was replaced daily. AAV was intravitreally injected after seven days of nicotine administration, and CNV was induced after three weeks. For the induction of CNV, mice were intraperitoneally anesthetized with pentobarbital. Then, 0.5% tropicamide and 0.5% phenylephrine eye drops were mixed and administered to dilute the pupils. Four laser burns were created around the optic nerve at a distance of 2 to 3 papillary diameters using an 810 nm laser (IRIDEX, Mountain View, CA, USA) with the following parameters: 200 mW power, 50 ms duration and 75 μm spot size. The appearance of a bubble during laser photocoagulation, which was considered to indicate the rupture of Bruch’s membrane, was taken as a sign of successful CNV induction.

### 4.12. Fundus Fluorescein Angiography (FFA)

FFA was used to record and assess the level of leakage using a retinal imaging system (Micron IV, PHOENIX, AZ, USA) seven days after laser photocoagulation. After anesthetization and pupil dilation, mice were intraperitoneally injected with fluorescein sodium (0.1%, 0.1 mL/10 g). FFA images were captured during the early phase (1–2 min after fluorescein injection) and the late phase (4–5 min after fluorescein injection). The CNV leakage grade was independently evaluated by two specialists. The criteria for grading FFA images included no hyperfluorescence (Grade 1); hyperfluorescence with no leakage (Grade 2); hyperfluorescence at the beginning and evident leakage at the late phase (Grade 3); and increasing evident leakage through the whole phase (Grade 4).

### 4.13. Choroidal Flat Mount and Immunostaining

Mice were sacrificed seven days after laser photocoagulation. The eyeballs of the mice were fixed in 4% paraformaldehyde for an hour and washed twice with PBS following enucleation. The remaining RPE-choroid-sclera complexes were made into flat mounts with four radial incisions and then blocked and permeabilized. Then, the tissues were washed and incubated with an anti-IB4 antibody at room temperature for 3 h. Then, the RPE-choroid-sclera complexes were washed, flat-mounted onto glass slides and covered with a cover slip. The immunostaining of flat mounts was captured and analyzed by confocal microscopy (Carl Zeiss, Oberkochen, Germany).

### 4.14. Statistical Analysis

All the measurements were performed in a blinded manner. The quantitative data are presented as the mean ± SEM. The differences between the two groups were analyzed by unpaired, two-tailed *t* test. When there were more than two groups, one-way ANOVA was performed. *p* values of less than 0.05 were considered statistically significant (* *p* < 0.05, ** *p* < 0.01, *** *p* < 0.001). GraphPad Prism 7 (GraphPad Software, San Diego, CA, USA) was used to conduct all the statistical analyses.

## Figures and Tables

**Figure 1 ijms-24-15523-f001:**
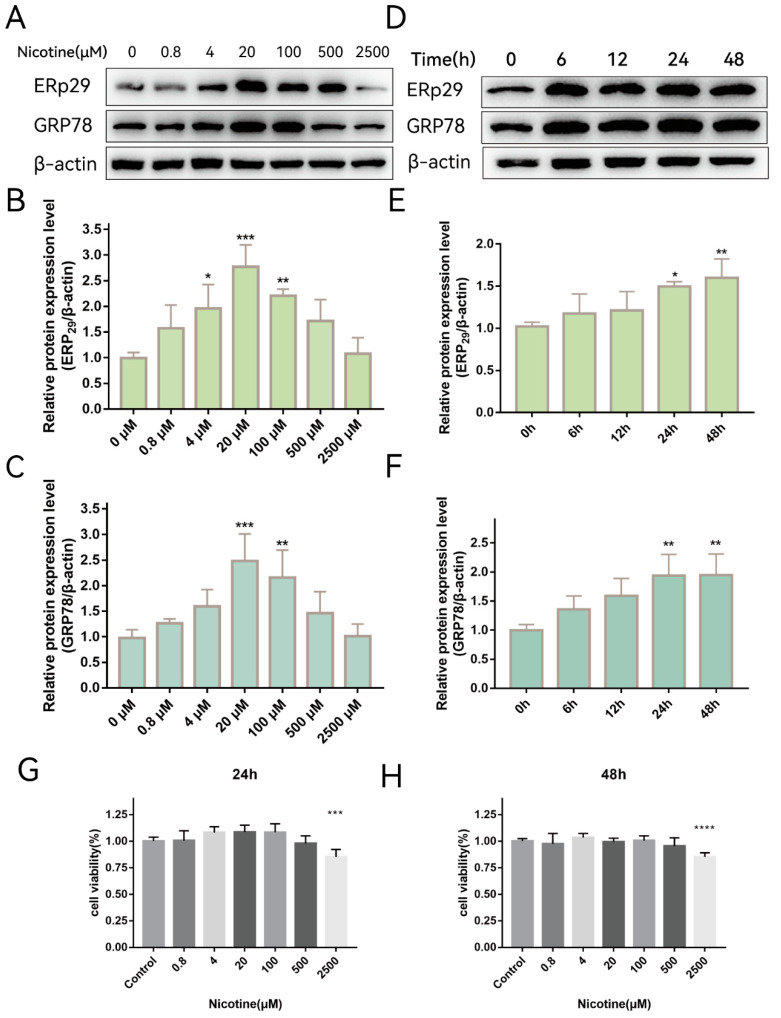
Nicotine Exacerbates ER Stress In Vitro. (**A**–**C**) Western blot showing the expression of ERp29 and GRP78 in ARPE-19 cells after exposure to different concentrations of nicotine for 24 h. (**D**–**F**) Western blot showing the expression of ERp29 and GRP78 at different times after treatment with 20 μM nicotine. (**G**,**H**) Statistical analysis of the CCK-8 assay shows ARPE-19 cell viability after exposure to different concentrations of nicotine for different times. Data are presented as the mean ± SEM of three independent experiments. * *p* < 0.05, ** *p* < 0.01, *** *p* < 0.001; **** *p* < 0.0001.

**Figure 2 ijms-24-15523-f002:**
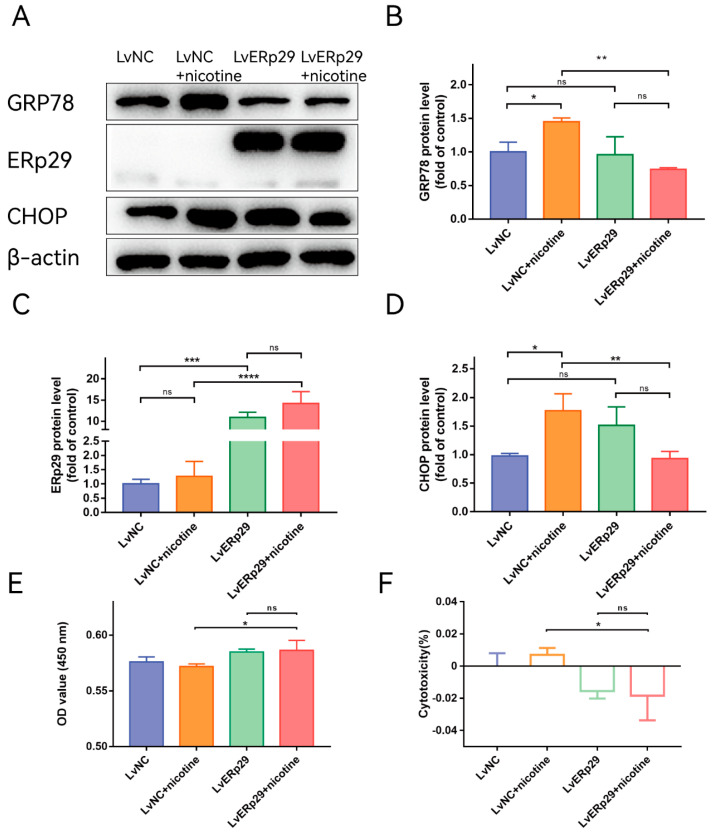
(**A**) Western blot showing the expression of GRP78, ERp29, and CHOP in each group. (**B**–**D**) The histogram shows the densitometric analysis of the average levels of GRP78, ERp29, and CHOP normalized to β-actin (n = 3). (**E**,**F**) CCK-8 assays demonstrated that overexpression of ERp29 reduced the cytotoxic effects of nicotine on ARPE-19 cells. The data are presented as the mean ± SEM. * *p* < 0.05, ** *p* < 0.01, *** *p* < 0.001; **** *p* < 0.0001. ns, no statistical significance.

**Figure 3 ijms-24-15523-f003:**
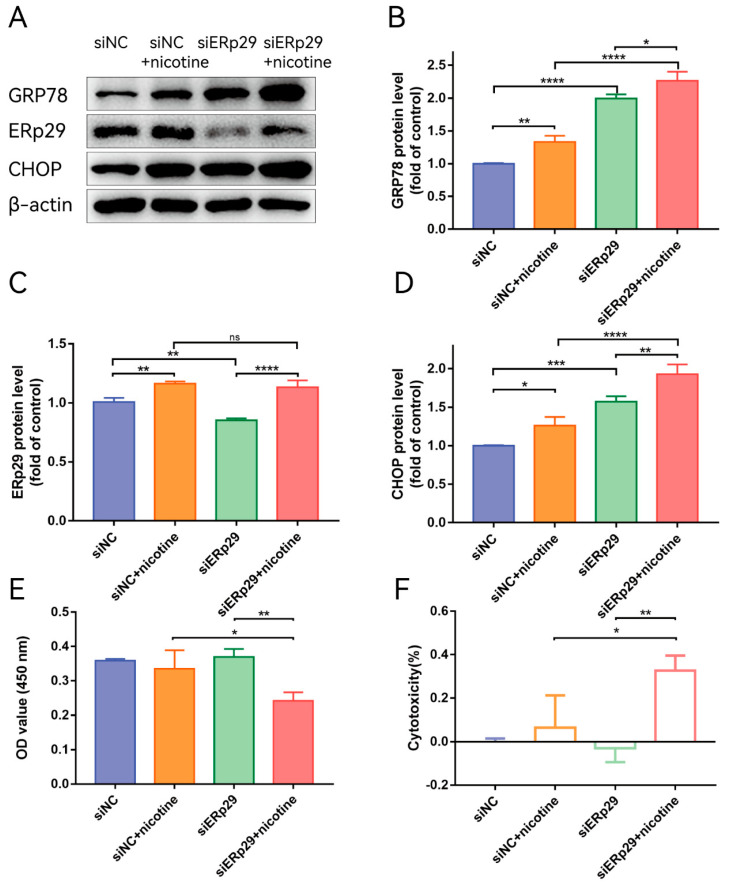
(**A**) Western blot showing the expression of GRP78, ERp29, and CHOP in each group. (**B**–**D**) The histogram shows the densitometric analysis of the average levels of GRP78, ERp29, and CHOP normalized to β-actin (n = 3). (**E**,**F**) CCK-8 assays demonstrated that ERp29 deficiency increased the cytotoxic effects of nicotine in ARPE-19 cells. The data are presented as the mean ± SEM. * *p* < 0.05, ** *p* < 0.01, *** *p* < 0.001; **** *p* < 0.0001. ns, no statistical significance.

**Figure 4 ijms-24-15523-f004:**
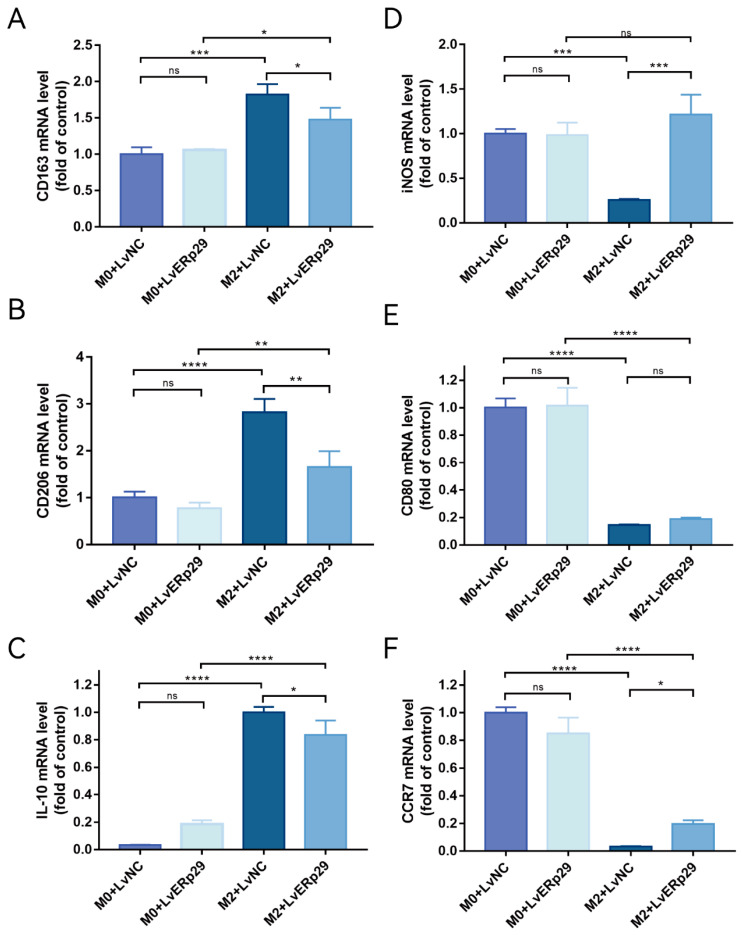
(**A**–**C**) qPCR showed the expression of the M2 macrophage markers CD163, CD206, and IL-10 in each group. (**D**–**F**) qPCR showed the expression of iNOS, CD80, and CCR7, which are M1 macrophage markers, in each group. The data are presented as the mean ± SEM of three independent experiments. * *p* < 0.05; ** *p* < 0.01; *** *p* < 0.001; **** *p* < 0.0001. ns, no statistical significance.

**Figure 5 ijms-24-15523-f005:**
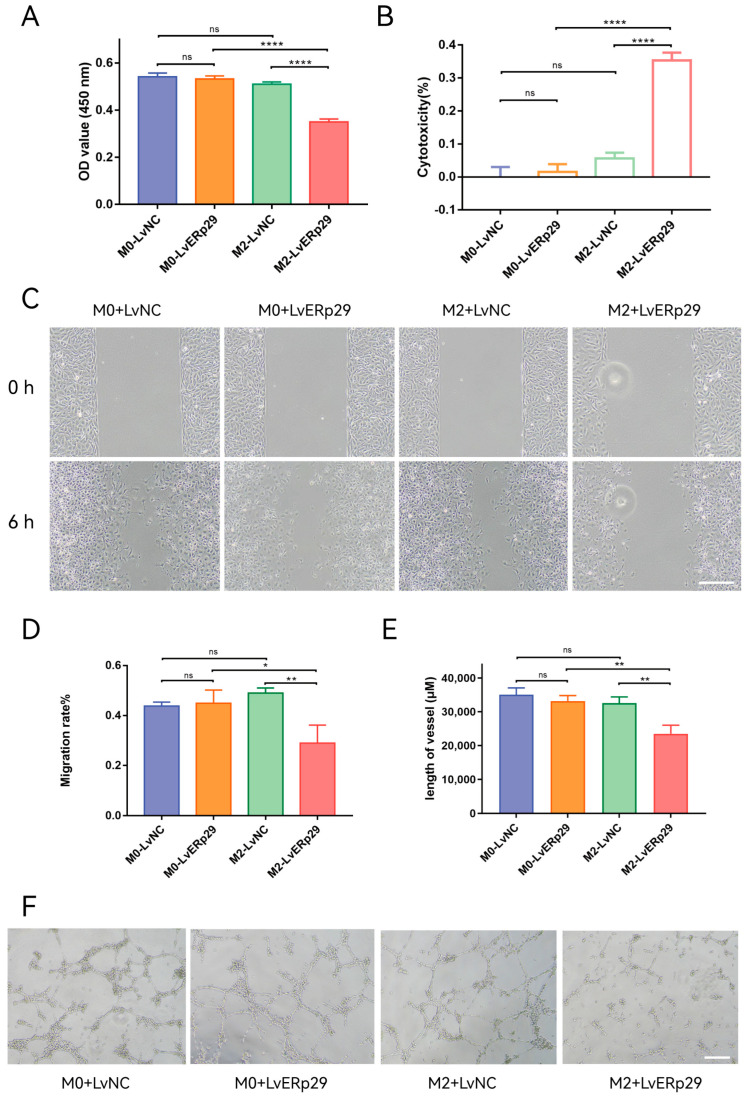
The effect of ERp29 on the viability, migration, and tube formation of HUVECs. (**A**,**B**) Cell viability analysis by CCK8 assay. HUVECs were treated with conditioned medium from different groups for 48 h. (**C**,**D**) Representative photomicrographs of each group at 0 h and 6 h in the wound-healing assay. Scale bar, 1 mm. (**E**,**F**) Representative photomicrographs of each group in the tube formation assay. The number of branches in each group was quantified. Scale bar, 1 mm. The data are presented as the mean ± SEM. * *p* < 0.05; ** *p* < 0.01; **** *p* < 0.0001. ns, no statistical significance.

**Figure 6 ijms-24-15523-f006:**
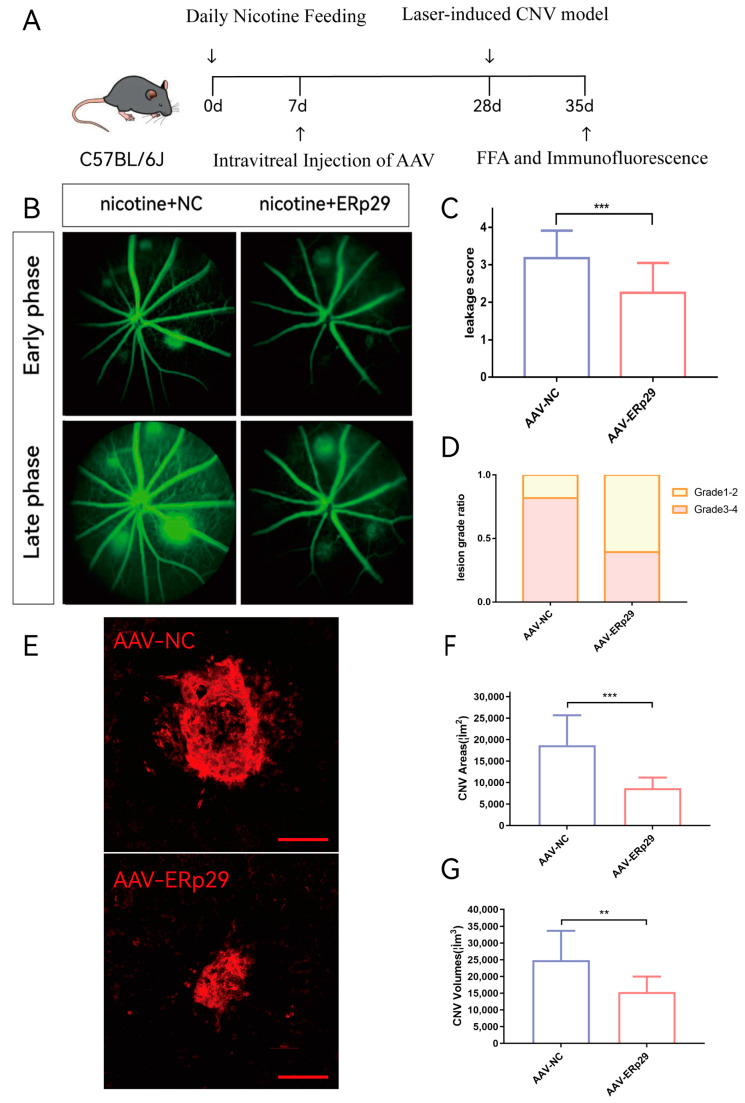
Observation of ERp29 overexpression in a mouse CNV model. (**A**) Mouse experiment process. (**B**–**D**) FFA images were captured in the early phase and late phase after fluorescein delivery one week after laser treatment. Vascular leakage was evaluated by determining average grades and grade distribution. The method of evaluation is described in the materials and methods. (**E**–**G**) The immunostaining of flat mounts was captured and analyzed by confocal microscopy. The CNV area and volume of were analyzed by ImageJ (ImageJ (https://imagej.nih.gov/ij/)). Schale bar, 100 μM. The data are presented as the mean ± SEM. ** *p* < 0.01; *** *p* < 0.001; ns, no statistical significance.

## Data Availability

The datasets generated during and/or analyzed during the current study are not publicly available due to the following study request but are available from the corresponding author on reasonable request.

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
