# Peer review of "ERp29 Attenuates Nicotine-Induced Endoplasmic Reticulum Stress and Inhibits Choroidal Neovascularization"

_ijms, 2023, doi:10.3390/ijms242115523_

Round 1
Reviewer 1 Report
I believe that the authors have successfully demonstrated that ERp29 overexpression has a down-regulation effect on macrophages (co-cultured with RPE cells), inhibits the viabilty, migration and tube formation of HUVECs and even has an in vivo effect of inhibiting the growth of CNV in mice.
Thereby, they have enhanced the existing level of evidence indicating that ERp29 may become a future tool in our fight against CNV in AMD patients.
I would like to ask the authors to elaborate on the fact that nicotine exposure increases the levels of ERp29 and GRP78, yet overexpression of ERp29 decreases the levels of GRP78. Is that because ERp29 and GRP78 play a similar role in ER homeostasis, as stated in lines 220-221?
line 200 - I do not believe that ImageJ can assess the volume of the CNV membrane, only the area.
When first referring to previous studies of the authors I believe that it would be helpful to insert the reference number. (line 78, line 98)
Also, please explain acronyms when they are first introduced in the text (e.g. CSE, CHOP, HUVEC).
Author Response
【1】I would like to ask the authors to elaborate on the fact that nicotine exposure increases the levels of ERp29 and GRP78, yet overexpression of ERp29 decreases the levels of GRP78. Is that because ERp29 and GRP78 play a similar role in ER homeostasis, as stated in lines 220-221?
Answer:Thank you for raising this question. Yes, ERp29 and GRP78 play a similar role in ER homeostasis. In the early stage of ER stress, the expression of GRP78 increases in the ER, which can maintain ER homeostasis by regulating unfolded protein reactions. In this respect,overexpression of ERp29, a molecular chaperone of GRP78, can decrease the levels of GRP78. ERp29 has two main functions under nicotine exposure as stated in lines 215-227.
【2】line 200 - I do not believe that Image J can assess the volume of the CNV membrane, only the area.
Answer:Thank you for raising this question. After making choroidal flat mount and immunostaining, we used the Z-stack mode of confocal microscope (Carl Zeiss, LSM980) to take fluorescence photos of different layers, and record them. When calculating the volume of CNV, we measure each image of the stack and calculate the volume by Image J using one method as below (Visikol® - Blog Post: Loading and Measurement of Volumes in 3D Confocal Image Stacks with ImageJ | Visikol). We have also provided you with two published articles using this method (PMID:32323368; PMID:32438838).
【3】When first referring to previous studies of the authors I believe that it would be helpful to insert the reference number. (line 78, line 98).Also, please explain acronyms when they are first introduced in the text (e.g. CSE, CHOP, HUVEC).
Answer:Thank you for this valuable feedback. We have added reference numbers and explanations for acronyms in the revised manuscript.

Reviewer 2 Report
Lu et al in their paper are describing the role of an ER-resident chaperone protein, endoplasmic reticulum protein 29 (ERp29), in nicotine-induced ER stress and choroidal neovascularization (CNV) that is a hallmark of wet age-related macular degeneration (wet AMD). AMD is a widespread retinal disorder which ultimately results in blindness. Along with genetic and environmental factors, smoking or exposure to smoke can be one of the major factors that might make a person susceptible to AMD. Therefore, understanding how smoking can lead to wet AMD will help find therapeutic targets to treat this disease.
Huang et al., have previously shown that ERp29 has a protective role in AMD. They showed that ERp29 reduces cigarette smoke extract induced- ER stress in retinal pigment epithelial cells (RPE) cells. It also helps RPE viability by maintaining RPE cell-cell tight junction proteins that maintain RPE integrity and health of the cells. Lu et al., in this paper are trying to study if ERp29 can have the same protective effect in the presence of nicotine, a component of cigarette smoke extract that was used by Huang et al., (2015). This paper confirms the protective role played by role ERp29 in cigarette smoke induced ER stress and goes on to further show that ERp29 can also reduce macrophage activation and inhibits CNV in vivo thereby preventing wet AMD progression.
The following are my comments.
1] Can you describe in lines 78-79, what is the difference between cigarette smoke extract and nicotine and why is it more relevant to study only the effect of nicotine?
2] If you compare Fig1F and 1G, levels of GRP78 are higher than ERp29. So, do you think GRP78 can also have a similar protective function in nicotine-induced ER stress, if you use overexpression and knockdown approach that you used for ERp29 protein?
3] In Fig3C, can you comment on the increase in ERp29 protein in the presence of nicotine in siERp29 treated cells?
4] Have you analyzed the condition media for expression of angiogenesis-related secreted proteins like VEGF, bFGF, PDGF, or EGF to understand proteins that ERp29 might be interacting with?
Author Response
【1】Can you describe in lines 78-79, what is the difference between cigarette smoke extract and nicotine and why is it more relevant to study only the effect of nicotine?
Answer:Thank you for raising this question. Nicotine is an important component in cigarette smoke extract and has multiple effects on the generation of CNV. Please refer to our background in lines 57-68 (Nicotine contributes to the formation and progression of CNV by regulating macrophage polarization, promoting RPE cell secretion of macrophage-related chemokines, recruiting and activating macrophages, and also inducing M2 macrophage polarization).
In summary, nicotine plays an important role in the impact of smoking on CNV. In order to further investigate the mechanism by which tobacco exacerbates CNV, we chose to focus on nicotine.
【2】If you compare Fig1F and 1G, levels of GRP78 are higher than ERp29. So, do you think GRP78 can also have a similar protective function in nicotine-induced ER stress, if you use overexpression and knockdown approach that you used for ERp29 protein?
Answer:Thank you for raising this question. Yes, we think GRP78 can also have a similar protective function in nicotine-induced ER stress. GRP78 can mitigate stress and maintain ER homeostasis. Since ERp29 is a molecular chaperone of GRP78, when we studied the role of ERp29 in endoplasmic reticulum stress, we simultaneously compared and observed the changes in GRP78. If necessary, please read the overview of GRP78. (PMID: 30978349)
【3】In Fig3C, can you comment on the increase in ERp29 protein in the presence of nicotine in siERp29 treated cells?
Answer:Thank you for this valuable feedback. Considering many functions of ERp29, we believe that ERp29 is an indispensable housekeeping gene for RPE cells. We downregulated the transcription level of ERp29 through siRNA, but after nicotine exposure, the RPE cells still upregulated the translation level of ERp29 to some extent. The regulation in different expression levels after nicotine exposure needs further study.
【4】 Have you analyzed the condition media for expression of angiogenesis-related secreted proteins like VEGF, bFGF, PDGF, or EGF to understand proteins that ERp29 might be interacting with?
Answer:Thank you for raising this question. No, our team have not analyzed the condition media for expression of angiogenesis-related secreted proteins yet. To further understand how ERp29 in RPE cells regulates macrophages and vascular endothelial cells in the inflammatory microenvironment, the secretion factors and extracellular vesicles in RPE cells should be analyzed in future. A stable monoclonal cell line that overexpresses ERp29 should be considered. For example, cells with random integration after lentivirus infection needs to be screened to find cells with normal cellular functions, single cell clones were used to establish some stable cell lines for subsequent cell interaction studies.
